# Causal Question Answering with Reinforcement Learning

## ABSTRACT

Causal questions inquire about causal relationships between different events or phenomena. Specifically, they often aim to determine whether there is a relationship between two phenomena, or to identify all causes/effects of a phenomenon. Causal questions are important for a variety of use cases, including virtual assistants and search engines. However, many current approaches to causal question answering cannot provide explanations or evidence for their answers. Hence, in this paper, we aim to answer causal questions with CauseNet, a large-scale dataset of causal relations and their provenance data. Inspired by recent, successful applications of reinforcement learning to knowledge graph tasks, such as link prediction and fact-checking, we explore the application of reinforcement learning on CauseNet for causal question answering. We introduce an Actor-Critic based agent which learns to search through the graph to answer causal questions. We bootstrap the agent with a supervised learning procedure to deal with large action spaces and sparse rewards. Our evaluation shows that the agent successfully prunes the search space to answer binary causal questions by visiting less than 30 nodes per question compared to over 3,000 nodes by a naive breadth-first search. Our ablation study indicates that our supervised learning strategy provides a strong foundation upon which our reinforcement learning agent improves. The paths returned by our agent explain the mechanisms by which a cause produces an effect. Moreover, for each edge on a path, CauseNet stores its original source on the web allowing for easy verification of paths.

## CCS CONCEPTS

• **Information systems** → **Question answering**; • **Computing methodologies** → *Reinforcement learning*; *Causal reasoning and diagnostics*; • **Mathematics of computing** → *Causal networks*.

## KEYWORDS

question answering, causality graphs, reinforcement learning

**ACM Reference Format:**
Anonymous Author(s). 2024. Causal Question Answering with Reinforcement Learning. In *Proceedings of the 2024 World Wide Web Conference (WWW '24), May 13–17, 2024, Singapore.* ACM, New York, NY, USA, 12 pages. https://doi.org/10.1145/nnnnnnn.nnnnnnn

## 1 INTRODUCTION

Causal question answering addresses the problem of determining the causal relations between given causes and effects [5, 19]. This involves examining *whether* a causal relation exists, *what* causal relations exist, and *how* causal relations can be explained in terms of intermediate steps. Examples of such questions include *"Does pneumonia cause anemia?"*, *"What are the effects of pneumonia?"*, and *"How does pneumonia cause death?"* Nowadays, the necessity

*WWW '24, May 13–17, 2024, Singapore*
2024. ACM ISBN 978-x-xxxx-xxxx-x/YY/MM…$15.00
https://doi.org/10.1145/nnnnnnn.nnnnnnn

to answer causal questions arises in various domains. For example, users often seek answers to causal questions from virtual assistants like Alexa or from search engines [13, 27]. Reasoning via chains of causal relations is crucial for argumentation [11, 44] and automated decision-making [12, 13, 19], too, e.g., to arrive at better answers and to gain a deeper understanding.

The literature started to introduce approaches for causal question answering [4, 12, 19, 37]. However, they lack explanations and verifiability of their answers due to a lack of large, high-quality datasets for causal relations [12, 19, 37]. Only recently, the introduction of CauseNet [13], a large-scale knowledge graph consisting of causal relations with context information, provides new opportunities to build effective, verifiable causal question answering systems that we exploit in this work.

Inspired by the successful application of reinforcement learning to knowledge graphs on different tasks such as link prediction [48], fact-checking [6], conversational question answering [18], and multi-hop reasoning [45], in this paper, we explore whether we can model the *causal question answering task* as a sequential decision problem over CauseNet. We train a reinforcement learning agent that learns to walk over CauseNet to find good inference paths to answer binary causal questions. We implement the agent via the Synchronous Advantage Actor-Critic (A2C) algorithm [26] and use generalized advantage estimation (GAE) [35] to compute the advantage. To address the challenge of a large action space in CauseNet [13], we bootstrap the agent with a supervised learning procedure [48] where the agent receives expert demonstrations to understand what good paths look like.

We evaluate our approach both on causal questions from SemEval [14] as well as a novel dataset constructed from the MS MARCO [13, 27] dataset, which consists of questions asked to search engines. While the former is skewed to questions with the answer "yes", the latter is balanced and contains an equal number of questions that are to be answered with "yes" or "no". Our evaluation demonstrates that on both datasets, our agent can effectively prune the search space, considering only a small number of nodes per question—on average, less than 30 nodes per question. For comparison, a breadth-first search (BFS) visits over 3,000 nodes per question. Furthermore, our agent minimizes false positives achieving a precision of 0.89, whereas BFS achieves a precision of only 0.75 and the language model UnifiedQA [20, 21] only a precision of 0.5. Our experiments confirm that bootstrapping the agent via supervised learning establishes a strong foundation decreasing uncertainty and accelerating the learning process. The paths found by our agent can be used to explain the relations between cause and effect, including the option to report the original source of the causal relation [13].

To summarize our contributions: (1) We introduce the first reinforcement learning approach for causal question answering on knowledge graphs; (2) we introduce a supervised learning procedure for causal question answering to handle the challenge of the large action space and sparse rewards and accelerate the learning process; (3) we introduce a new causal question dataset.

## 2 RELATED WORK

In the following, we summarize related work regarding causal knowledge graphs, approaches for causal question answering, and approaches that apply reinforcement learning to reasoning tasks on knowledge graphs.

*Causal Knowledge Graphs.* ConceptNet [41] is a general knowledge graph (KG) consisting of 36 relations between natural language terms, including a *Causes* relation. CauseNet [13] and Cause Effect Graph [22] specifically focus on causal relations extracted via linguistic patterns from web sources like Wikipedia and ClueWeb12.[1] ATOMIC [34] focuses on inferential knowledge of commonsense reasoning in everyday life. It consists of *"If-Event-Then-X"* relations based on social interactions or real-world events. $ATOMIC_{20}^{20}$ [16] selects relations from ATOMIC and ConceptNet to create an improved graph while adding more relations via crowdsourcing. Instead, West et al. [46] automate the curation of causal relations by clever prompting of a language model. Finally, CSKG [17] builds a consolidated graph combining seven knowledge graphs, including ConceptNet and ATOMIC. We use CauseNet because most of the other KGs are smaller and less focused on causal relations, e.g., CauseNet contains many more causal relationships than ConceptNet [13]. Future work may involve combining multiple KGs.

*Causal Question Answering.* As of now, only few approaches tackle the causal question answering task. Most of them focus on binary questions, i.e., questions such as *"Does X cause Y?"* which expect a "yes" or "no" answer. Kayesh et al. [19] model the task as a transfer learning approach. They extract cause-effect pairs from news articles via causal cue words. Subsequently, they transform the pairs into sentences of the form *"X may cause Y"* and use them to finetune BERT [8]. Similarly, Hassanzadeh et al. [12] employ large-scale text mining to answer binary causal questions introducing multiple unsupervised approaches ranging from string matching to embeddings computed via BERT. Sharp et al. [37] consider multiple-choice questions of the form *"What causes X?"*. First, they mine cause-effect pairs from Wikipedia via syntactic patterns and train an embedding model to capture the semantics between them. At inference time, they compute the embedding similarity between the question and each answer candidate. Dalal [4], Dalal et al. [5] combine a language model with CauseNet [13]. Given a question, they apply string matching to extract relevant causal relations from CauseNet. Subsequently, they provide the question with the causal relations as additional context to a language model. As other language model-based approaches, too, they cannot produce verifiable answers. None of them selects relevant paths in a causality graph via reinforcement learning.

*Knowledge Graph Reasoning with Reinforcement Learning.* In recent years, reinforcement learning on knowledge graphs has been successfully applied to link prediction [6], fact-checking [48], or question answering [31]. Given a source and a target entity, DeepPath [48] learns to find paths between them. The training of DeepPath involves two steps. First, it is trained via supervised learning and afterward via REINFORCE [47] policy gradients. During inference time, the paths are used to predict links between entities

---

[1] https://lemurproject.org/clueweb12/

or check the validity of triples. Subsequently, MINERVA [6] improves on DeepPath by introducing an LSTM [15] into the policy network to account for the path history. Moreover, MINERVA does not require knowledge of the target entity and is trained end-to-end without supervision at the start. Lin et al. [23] propose two improvements for MINERVA. First, they apply reward shaping by scoring the paths with a pre-trained KG embedding model [7] to reduce the problem of sparse rewards. Second, they introduce a technique called action dropout, which randomly disables edges at each step. Action dropout serves as additional regularization and helps the agent learn diverse paths. M-Walk [38] applies model-based reinforcement learning techniques. Like AlphaZero [40], M-Walk applies Monte Carlo Tree Search (MCTS) as a policy improvement operator. Thus, at each step, M-Walk applies MCTS to produce trajectories of an improved policy and subsequently trains the current policy to imitate the improved one. GaussianPath [45] takes a Bayesian view of the problem and represents each entity by a Gaussian distribution to better model uncertainty. Previous approaches for reinforcement learning on knowledge graphs [6, 31, 48] were designed for knowledge graphs with multiple relation types which enabled them to use the relations types as actions at each time step. In contrast, we tailor our approach to causal knowledge graphs that contain only one relation type "cause". Thus, the relations do not provide any learning signal, which makes the question answering task particularly challenging. In our case, the action space consists of all entities in the knowledge graph[2] and the actions the agent can take change at each time step, due to the different neighborhoods of each entity. Moreover, we employ an actor-critic reinforcement learning algorithm to boost performance.

## 3 CAUSAL QUESTION ANSWERING WITH REINFORCEMENT LEARNING

In the following, we formulate the question-answering task as a sequential decision problem on a causal knowledge graph and define the environment of the reinforcement learning agent. Afterward, we present our reinforcement learning agent, including the network architecture, the training procedure, the search strategy at inference time, and an approach to bootstrapping the agent with supervised learning.

### 3.1 Problem Definition

Given a causal question $q$ in natural language and a causal knowledge graph $\mathcal{K} = \{(h, r, t)\} \subseteq \mathcal{E} \times \mathcal{R} \times \mathcal{E}$, where $h, t \in \mathcal{E}$ denote entities and $r \in \mathcal{R}$ denotes a relation, the agent walks over the graph to answer the question. Note that the causal knowledge graph considered in this paper only contains the *cause* relation such that $\mathcal{R} = \{cause\}$. We consider binary causal questions $q$, where the agent has to determine the validity of a causal relation such as *"Can X cause Y?"* where $X$ and $Y$ represent a cause and effect, respectively.

In the following, we elucidate the binary causal question answering task on the knowledge graph $\mathcal{K}$ on the basis of the example in Figure 1. The example shows an excerpt of a causal knowledge graph (CauseNet) and the binary causal question *"Does pneumonia cause anemia?"*. In this question, *pneumonia* takes the role of the

---

[2] In case of CausNet, we have 80,222 entities [13].

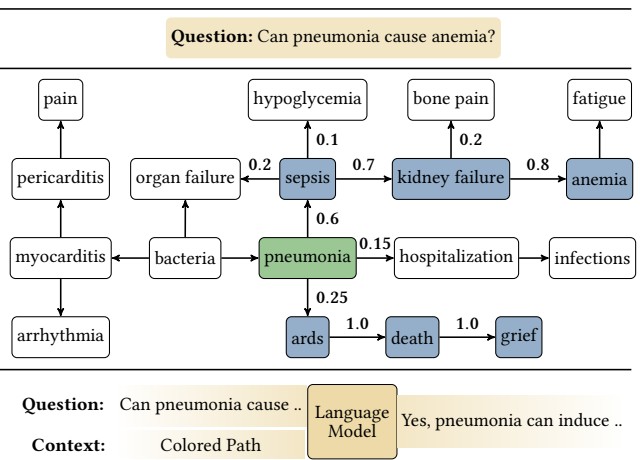

**Figure 1: An excerpt from CauseNet [13] showing the entity *pneumonia* together with its neighborhood containing causes and effects, where each edge depicts a *cause* relation. The numbers on the edges show the probability of taking this edge under the current policy $\pi_\theta(a_t|s_t)$. For brevity, we only show the relevant probabilities for the given paths. The lower part of the figure shows the possibility to combine our agent with a language model. In that setup, we provide the paths the agent learned as additional context to the language model.**

cause, and *anemia* the role of the effect. First, the cause and effect are linked to the graph. Therefore, we find entities $e_c, e_e \in \mathcal{E}$ such that *pneumonia* maps to $e_c$ and *anemia* to $e_e$. Currently, we link them via exact string matching. However, more sophisticated strategies can be considered in future work [18]. Consequently, starting from $e_c$, the agent has to find a path $(e_c, e_1, e_2, \ldots, e_e)$ with $e_i \in \mathcal{E}$, where the agent arrives at the effect $e_e$.[3] If the agent finds such a path, the question is answered with "yes" and otherwise with "no". For the example, a possible path is *(pneumonia, sepsis, kidney failure, anemia)*. Afterward, we can inspect the path to get further insights into the relationship between cause and effect.

## 3.2 Environment

As done by related work [6, 31, 48], we formulate the causal question answering task as a sequential decision problem on the knowledge graph $\mathcal{K}$. The agent walks over the graph and decides which edge to take at each entity. Therefore, we define a Markov Decision Process (MDP) as a 4-tuple $(\mathcal{S}, \mathcal{A}, \delta, \mathcal{R})$. The MDP consists of the state space $\mathcal{S}$, the action space $\mathcal{A}$, the transition function $\delta : \mathcal{S} \times \mathcal{A} \rightarrow \mathcal{S}$, and the reward function $\mathcal{R} : \mathcal{S} \rightarrow \mathbb{R}$. Hence, at each state $s_t \in \mathcal{S}$ the agent selects an action $a_t \in \mathcal{A}$, which changes the current state via $\delta(s_t, a_t)$ to $s_{t+1}$. Additionally, the agent receives a reward $\mathcal{R}(s_{t+1}) = r_t$. Note that the transition function $\delta$ is known and deterministic because the graph entirely defines $\delta$. So, for each action $a_t$ in state $s_t$, the next state $s_{t+1}$ is known.

---

[3]The path only shows the entities, because the graph contains only one relation type.

*Agent.* Our agent consists of a policy network $\pi_\theta(a_t|s_t)$ (Actor) parameterized with $\theta$ and a value network $V_\psi(s_t)$ (Critic) parameterized with $\psi$. The policy network $\pi_\theta(a_t|s_t)$ generates a distribution over actions $a_t$ at the current state $s_t$. The value network $V_\psi(s_t)$ generates a scalar to estimate the value of the state $s_t$. Specifically, the value network should predict the future reward from $s_t$ onwards.

*States.* At each time step t, we define state $s_t = (\mathbf{q}, e_t, \mathbf{e_t}, \mathbf{h_t}, e_e) \in \mathcal{S}$, where $\mathbf{q}$ represents the embedding of the question $q$, $e_t \in \mathcal{E}$ the current entity, and $\mathbf{e_t}$ its embedding. The entity $e_t$ is needed to define the action space, and its embedding $\mathbf{e_t}$ is used as input to the agent's networks. Additionally, $\mathbf{h_t}$ represents the path history of the agent and $e_e$ the entity corresponding to the effect found in the question $q$. Moreover, $e_0 = e_c$ and $\mathbf{h_0} = \mathbf{0}$, where $e_c$ is the entity corresponding to the cause of the question (e.g., *pneumonia* in the example in Figure 1). The path history is represented by the hidden states of an LSTM.

*Actions.* The action space at each time step $t$ consists of all neighboring entities of the current entity in state $s_t$. Therefore, the set of possible actions in state $s_t = (\mathbf{q}, e_t, \mathbf{e_t}, \mathbf{h_t}, e_e)$ is defined as $A(s_t) = \{e|(e_t, r, e) \in \mathcal{K}\}$ where $r = cause$. So only the current entity $e_t$ is needed to define the action space $A(s_t)$. Note that while the additional components inside the state are not needed to define the action space, they are needed for other parts of the learning algorithm, as described below in Sections 3.3 and 3.4.

CauseNet [13] contains additional meta-information for each relation in the form of the original sentence $s$ from which the relation was extracted. Therefore, we include the original sentence $s$ when computing the embedding $\mathbf{a_t}$ for an action $a_t$. This is done by concatenating the sentence embedding $\mathbf{s}$ with the embedding of the entity $e = a_t$. Thus, the action embedding becomes $\mathbf{a_t} = [\mathbf{s}; \mathbf{e}]$.

As done by prior works [6, 23, 31], we add a special *STAY* action at each step, so the action space becomes $A(s_t) = A(s_t) \cup \{STAY\}$. When selecting this action, the agent stays at the current entity. This way, we can keep all episodes to the same length, even though different questions might require a different number of hops. Another option would be to add a stop action. However, in that case, we would have episodes of different lengths.[4] Moreover, we add inverse edges to the graph because our experiments showed that their addition increases the performance. In general, inverse edges allow the agent to undo wrong decisions and to reach nodes that could otherwise not be reached under a given episode length. We discuss some implications and tradeoffs of inverse edges in Section 5.

*Transitions.* As described above, the transition function is deterministic, so the next state is fixed after the agent selects an action. Let $s_t = (\mathbf{q}, e_t, \mathbf{e_t}, \mathbf{h_t}, e_e)$ be the current state and $a_t \in A(s_t)$ be the selected action in $s_t$. Subsequently, the environment evolves via $\delta(s_t, a_t)$ to $s_{t+1} = (\mathbf{q}, e_{t+1}, \mathbf{e_{t+1}}, \mathbf{h_{t+1}}, e_e)$, where $e_{t+1} = a_t$.

*Rewards.* The agent only receives a terminal reward at the final time step $T - 1$. Specifically, the agent receives a reward of $\mathcal{R}(s_{T-1}) = 1$ if $s_{T-1} = (\mathbf{q}, e_{T-1}, \mathbf{e_{T-1}}, \mathbf{h_{T-1}}, e_e)$ with $e_{T-1} = e_e$. Conversely, the agent receives a reward of $\mathcal{R}(s_{T-1}) = 0$ if $e_{T-1} \neq e_e$. Similarly, for all other time steps $t < T - 1$ the reward is 0 as well.

---

[4]In principle, episodes of different lengths are not a problem. However, keeping them to the same length simplifies the implementation. We chose this simplification because it worked well in prior works [6, 31].

*Path Rollouts — Episodes.* We define a path rollout or episode as a sequence of three tuples containing a state, action, and reward. Assuming a path rollout length of $T$, an example for a path rollout is: $((s_0, a_0, r_0), \ldots, (s_{T-2}, a_{T-2}, r_{T-2}))$. For path rollout length $T$, a path rollout contains $T - 1$ tuples. This is because the last state $s_{T-1}$ is only needed for the calculation of reward $r_{T-2}$, and no further action is taken.[5] For brevity, the rewards $r_t$ can be omitted.

## 3.3 Network Architecture

We use a Long Short-Term Memory (LSTM) [15] to parametrize our agent. Additionally, we experimented with a simple feedforward architecture but found that incorporating the path history is crucial for our needs. This aligns with previous research, where approaches such as MINERVA [6] and SRN [31] also used LSTMs and GRUs. Just CONQUER [18] used a feedforward architecture, but they only considered paths of length one.

Let $\mathbf{q} \in \mathbb{R}^d$ be the embedding of the question $q$ and $\mathbf{E} \in \mathbb{R}^{|\mathcal{E}| \times d}$ the embedding matrix containing the embeddings for each entity $e \in \mathcal{E}$ of the knowledge graph $\mathcal{K}$. The parameter $d$ specifies the dimension of the embeddings. The LSTM is then applied as

$$\mathbf{h_t} = \begin{cases} LSTM(\mathbf{0}; [\mathbf{q}, \mathbf{e_c}]), & \text{if } t = 0 \\ LSTM(\mathbf{h_{t-1}}, [\mathbf{q}; \mathbf{e_t}]), & \text{otherwise} \end{cases} \quad (1)$$

where $\mathbf{h_t} \in \mathbb{R}^{2d}$ represents the hidden state vector (history) of the LSTM, and $[;]$ is the vector concatenation operator. At each time step, the LSTM takes the previous history $\mathbf{h_{t-1}}$ and the concatenation of the question embedding $\mathbf{q}$ and the current node embedding $\mathbf{e_t} \in \mathbb{R}^d$ to produce $\mathbf{h_t}$. In the first time step, $\mathbf{h_0}$ is initialized with the zero vector and $\mathbf{e_0} = \mathbf{e_c}$, where $\mathbf{e_c}$ is the embedding of the entity corresponding to the cause found in the current question.

On top of the LSTM, we stack two feedforward networks: one for the policy network $\pi_\theta(a_t|s_t)$ and one for the value network $V_\psi(s_t)$. In Section 3.2, we defined the action space $\mathcal{A}(s_t)$ at time step $t$ and state $s_t = (\mathbf{q}, e_t, \mathbf{e_t}, \mathbf{h_t}, e_e)$ to contain all neighbors of the entity $e_t$. Therefore, we introduce an embedding matrix $\mathbf{A_t} \in \mathbb{R}^{|A(s_t)| \times 2d}$, where the rows contain the embeddings of the actions $a_t \in \mathcal{A}(s_t)$. The output of the policy network $\pi_\theta(a_t|s_t)$ is computed as

$$\pi_\theta(a_t|s_t) = \sigma(\mathbf{A_t} \times W_2 \times ReLU(W_1 \times \mathbf{h_t}))$$
$$a_t \sim Categorical(\pi_\theta(a_t|s_t)) \quad (2)$$

where $W_1 \in \mathbb{R}^{h \times 2d}$ and $W_2 \in \mathbb{R}^{2d \times h}$ are weight matrices with hidden dimension $h$ and $\sigma$ is the softmax operator. The final output of the policy network is a categorical probability distribution over all actions $a_t \in \mathbb{A}(s_t)$. Similarly, the output of the value network $V_\psi(s_t)$ is computed with the feedforward network

$$V_\psi(s_t) = W_4 \times ReLU(W_3 \times \mathbf{h_t}) \quad (3)$$

where $W_3 \in \mathbb{R}^{h \times 2d}$ and $W_4 \in \mathbb{R}^{1 \times h}$ are weight matrices with hidden dimension $h$, and the output is a scalar that estimates the future reward from state $s_t$ onwards. Overall, the weights of the LSTM are shared between the policy and value network, while each network has its own weights in the form of its feedforward head.

---

[5]See Algorithm 1 Lines 17-22 for more details.

## 3.4 Training the Reinforcement Learning Agent

The training process involves pre-processing questions and linking them to CauseNet entities. As CauseNet [13] does not contain negative information (see Section 7.1 in appendix), we only train the agent on positive causal questions, i.e., questions whose answer is "yes". We remove all questions where the cause, effect, or both cannot be found in CauseNet. Then we obtain embeddings for the entities from their textual representation, initialize agent weights, and sample path rollouts with the policy network. The training utilizes the Synchronous Advantage Actor-Critic (A2C) algorithm, with the policy network acting as the actor and the value network as the critic. The policy network update rule includes the generalized advantage estimate (GAE). As commonly done, we add an entropy regularization term to help the agent with the exploitation vs. exploration tradeoff. Simultaneously, the value network is updated. Further details on our training procedure for the policy network including the pseudocode can be found in Section 7.3 in the appendix.

## 3.5 Search Strategy

At inference time, the agent receives both positive and negative questions. To answer a given question, we sample multiple paths $p$ of length $T$ from the agent. If any path contains the entity $e_e$, the agent answers the question with "Yes", otherwise with "No". In case the cause, effect, or both cannot be found in CauseNet, the question is answered with "no" per default.

For each path rollout $((s_0, a_0), (s_1, a_1), \ldots, (s_{T-2}, a_{T-2}))$, the path that was taken on the graph consists of the entity $e_0$ in $s_0$ and the actions taken at each time step $t$, i.e., $p = (e_0, e_1, \ldots, e_{T-1})$ where $a_{t-1} = e_t \in \mathcal{E}$ for $t > 0$. The probability of path $p$ is the product

$$\mathbb{P}(p) = \prod_{t=0}^{T-2} \pi_\theta(a_t|s_t) \quad (4)$$

of the probabilities of taking action $a_t$ at state $s_t$ under the current policy $\pi_\theta(a_t|s_t)$ for $t \in \{0, \ldots, T-2\}$. Figure 1 shows an excerpt from CauseNet where each edge is annotated with the probability of taking this edge under the current policy.

To sample paths from the agent, we apply greedy decoding or beam search. *Greedy decoding* takes the action with the highest probability at each time step, i.e., $arg\,max_{a_t \in \mathcal{A}(s_t)} \pi_\theta(a_t|s_t)$. In Figure 1, the agent would select *sepsis* in the first time step, *kidney failure* in the second time step, and *anemia* in the third time step. However, one disadvantage of greedy decoding is its myopic behavior, i.e., it might miss high-probability actions in later time steps. *Beam search* tries to alleviate this problem by always keeping a set of the best partial solutions up to the current timestep. In our case, partial solutions are paths of length $t$, where $t$ is the current timestep. Furthermore, the paths are ranked by their probability, as defined in Equation 4. Assuming a beam width of two, Figure 1 shows the two paths that are found by beam search for the example. After obtaining these paths, the agent examines each one to determine whether it includes the effect. If that is the case, the agent answers the question with "yes", otherwise with "no". In this example, *anemia* is found, so the agent answers with "yes".

**Table 1: Number of questions for training, validation, and testing for the MS MARCO [27] and SemEval [37] datasets. The "E. Train" column shows the number of questions the agent has effectively available for learning.**

| Dataset | Train | | Validation | | Test | | E. Train |
|---|---|---|---|---|---|---|---|
| | Pos. | Neg. | Pos. | Neg. | Pos. | Neg. | Pos. |
| MS MARCO | 1837 | 332 | 194 | 47 | 223 | 40 | 1350 |
| SemEval | 694 | 690 | 84 | 89 | 87 | 86 | 812 |

## 3.6 Bootstrapping via Supervised Learning

Reinforcement learning algorithms often take a long time to converge due to their trial-and-error nature combined with large action spaces and sparse rewards [23, 48]. Thus, the reinforcement learning agent can be bootstrapped, by first training it on a series of expert demonstrations. For example, AlphaGo [39] trained the agent on demonstrations from expert Go players before continuing with their reinforcement learning algorithm. In our case, the expert demonstrations come from a breadth-first search (BFS) on CauseNet. First, we randomly select a subset $\overline{Q}$ of size $\alpha \cdot |questions|$ of the training questions, where $\alpha$ is a hyperparameter. Subsequently, we run a BFS on the cause $e_c$ and effect $e_e$ of each question in $\overline{Q}$ and build a path rollout for each found path. If a path rollout is shorter than the path rollout length $T$, it is padded with the *STAY* action.[6] Next, we train the policy network $\pi_\theta(a_t|s_t)$ via REINFORCE

$$\nabla_\theta J(\theta) = -\frac{1}{B} \sum_{i}^{B} \sum_{t=0}^{T-2} \nabla_\theta \log(\pi_\theta(a_t|s_t)) \, r_t + \beta H_{\pi_\theta} \qquad (5)$$

where $B$ is the batch size, $T$ the path rollout length, and $H_{\pi_\theta}$ the entropy regularization from Section 3.4. During supervised training, the reward $r_t$ is set to 1 at each step. Note that we only train the policy network $\pi_\theta(a_t|s_t)$ during supervised learning. Afterward, we further train both the policy and value networks via Algorithm 1, as explained in Section 3.4.

## 4 EVALUATION

In this section, we present the evaluation of our approach. We start by providing an overview of the experimental setup, including the datasets, the baselines, evaluation measures, hyperparameter settings, and implementation details. Afterward, we compare our agent to two baselines on the binary causal question answering task. Next, we conduct an ablation analysis to evaluate the effectiveness of the different components of our approach and evaluate the effects of initial supervised learning. Finally, we provide a few example paths found by our agent. The code and data to reproduce our results is publicly available.[7]

### 4.1 Experimental Setup

*Datasets.* As datasets, we employ subsets of causal questions from MS MARCO [27] and SemEval [14, 37]. To extract binary

---

[6]A question $q$ is discarded, if a path of length less than or equal to $T$ cannot be found between its cause $e_c$ and its effect $e_e$.

[7]https://github.com/CausalRLQA/CausalRLQA

causal question from MS MARCO we extended an extraction mechanism from Heindorf et al. [13] by including additional causal cue words [10]. SemEval was curated by Sharp et al. [37] by selecting a subset of 1730 word pairs from the semantic relation classification benchmark SemEval 2010 Task 8 [14]. Among the 1730 word pairs, there are 865 causal pairs and 865 non-causal pairs, i.e., the dataset is balanced, whereas MS MARCO is imbalanced. The first three columns of Table 1 show the original numbers of training, validation, and test questions of both datasets. However, as discussed in Sections 3.4 and 7.3, for training our reinforcement learning agent, we remove negative questions and questions where either the cause or effect cannot be found in CauseNet. The "E. Train" column shows the number of questions effectively available for learning when combining the training and validation sets. That leaves us with 1350 questions for MS MARCO and 812 questions for SemEval. For testing question answering, we use both positive and negative questions regardless of whether the cause/effect can be mapped to CauseNet. If a cause or effect cannot be found in CauseNet, the question is answered with "No". Further details regarding Causenet and our dataset construction are given in Sections 7.1 and 7.2 in the appendix, respectively.

*Baselines.* We compare our agent with two baselines: a breadth-first search (BFS) on CauseNet and the question answering system UnifiedQA-v2 [20, 21]. BFS performs an exhaustive search in the graph up to a certain depth and serves as a strong baseline. However, it must be noted that BFS can be applied effectively only to binary causal questions. Moreover, it needs to traverse many nodes in the graph whereas our reinforcement learning agent visits much fewer nodes. As a second baseline, we use UnifiedQA-v2 [20, 21]. UnifiedQA-v2 is a text-to-text language model based on the T5 architecture [32] and achieved state-of-the-art performance on multiple datasets. We chose UnifiedQA-v2 because it was used by the CausalQA [3] benchmark for their evaluation. Its input consist of a question and additional contextual information as shown in Figure 1. We experimented with three variants of UnifiedQA-v2: (1) with an empty context (UnifiedQA-v2), (2) with causal triples as context (UnifiedQA-v2-T) as done by [4, 5], (3) by using the provenance data available in CauseNet along paths from the cause to the effect (UnifiedQA-v2-P). For (2), all triples from CauseNet are obtained where the cause in the question matches the cause in CauseNet and the effect in the question matches the effect in CauseNet. For (3), we take advantage of the additional meta-information available in CauseNet [13]. For each causal pair, CauseNet contains the original sentence from which the pair was extracted. Given a question $q$ and a path $(e_1, \ldots, e_n)$ our agent found for that question, we extract the original sentence for each causal pair $(e_i, e_{i+1})$, with $0 \leq i < n$, on the path $p$. We concatenate the sentences for all paths and all pairs therein and input the sequence into the language model as context.

*Evaluation Measures.* We evaluate our agent for binary question answering using the standard classification measures accuracy, $F_1$-score, precision, and recall. Additionally, we evaluate our agent and the BFS baselines w.r.t. the number of unique nodes (entities) that are visited per question on average.

*Hyperparameter Optimization.* We optimized hyperparameters using Optuna [1] on the validation sets and subsequently retrained on the combined training and validation sets. We trained each agent for 2000 steps with a batch size of 128 and a learning rate of 0.0001. The hidden dimension of the feedforward heads was set to 2048. Additionally, we set the discount factor $\gamma$ to 0.99 and the $\lambda$ parameter of GAE to 0.95 [35]. The weight $\beta$ for the entropy regularization was set to 0.01. During supervised learning we used 300 training steps, a batch size of 64, and a supervised ratio $\alpha$ of 0.8. Moreover, we used a beam width of 50.

*Implementation Details.* We used the AdamW [24] optimizer with gradient norm clipping [28] at a value of 0.5. As knowledge graph we used CauseNet-Precision [13] and to embed the entities and questions we used GloVe embeddings [30]. For the UnifiedQA-v2 baseline we chose the *base* model [20, 21]. All experiments were run on a NVIDIA A100 40GB. Additional details can be found in our GitHub repository (URL see above).

## 4.2   Evaluation of the RL Agent

The last column of Table 2, compares the number of visited nodes by our approach with a brute-force BFS. Our approach visits less than 30 nodes per question on average whereas a BFS visits over 3,000 nodes to answer binary questions with 3 or 4 hops. Thus, our approach effectively prunes the search space by 99%. The fact that the number of nodes visited by BFS barely increases from 3 to 4 hops can be attributed to the topology of CauseNet.

The first columns of Table 2 compare the question-answering performance of our agent with the BFS and the language model UnifiedQA. We can observe that our agent achieves better precision for all configurations. In terms of F1 measure, our agent comes close to the performance of the BFS and UnifiedQA. On SemEval, the results of BFS are worse in the 3-Hop setting compared to the 2-Hop setting due to the introduction of many false positives through inverse edges.[8] As we discuss in 5, inverse edges can lead to mistakes by potentially introducing false positives. When going from 2 hops to 3 hops on SemEval, it is possible to reach more false positives than before. This is especially indicated by the reduction of precision from 0.937 to 0.750. In contrast, our agent mitigates this problem by pruning the search space and avoiding paths leading to wrong answers, still achieving a precision of 0.929. Compared to BFS, our agent has several advantages: (1) it prunes the search space and decreases the number of visited nodes by around 99%, (2) it can avoid false positives introduced by inverse edges and errors in CauseNet as shown above, (3) it can be extended to open-ended causal questions as discussed in Section 5.

Comparing our approach to a language model such as UnifiedQA, we observe that we consistently achieve a high precision above or around 0.9 whereas UnifiedQA only achieves a precision around 0.5. We attribute this to the tendency of UnifiedQA-v2 to guess and answer with "yes" most of the time. This works relatively well on MS MARCO dataset that is skewed towards positive questions (85% positive, 15% negative) and less so on SemEval that is balanced at

---

[8]The results of BFS 3-Hop on SemEval are improved when not using inverse edges. However, using inverse edges improves the results for all other configurations. Thus, we included them in the graph as they also improve the performance of our agent, as shown in the ablation study in Section 4.3.

**Table 2: Evaluation results of our agent on the MS MARCO and SemEval test sets compared to the BFS baseline and UnifiedQA-v2 baseline. The table reports the accuracy: A, $F_1$-Score: $F_1$, recall: R, precision: P, and the average number of nodes that were visited per question |Nodes|. The results for UnifiedQA-v2 using triples or paths of the agent as context are denoted as *UnifiedQA-v2-T* and *UnifiedQA-v2-P*.**

| MS MARCO | | | | |
| --- | --- | --- | --- | --- |
| | **A** | **$F_1$** | **R** | **P** | **|Nodes|** |
| BFS 1-Hop | 0.259 | 0.241 | 0.139 | 0.912 | 56.98 |
| BFS 2-Hop | 0.494 | 0.612 | 0.471 | 0.875 | 1726.71 |
| BFS 3-Hop | 0.589 | 0.714 | 0.605 | 0.871 | 3338.75 |
| BFS 4-Hop | 0.612 | 0.734 | 0.632 | 0.876 | 3494.94 |
| UnifiedQA-v2 | 0.722 | 0.828 | 0.789 | 0.871 | – |
| UnifiedQA-v2-T | 0.741 | 0.843 | 0.821 | 0.867 | – |
| UnifiedQA-v2-P | 0.661 | 0.789 | 0.740 | 0.842 | – |
| Agent 1-Hop | 0.255 | 0.234 | 0.135 | 0.909 | 14.02 |
| Agent 2-Hop | 0.460 | 0.562 | 0.408 | 0.901 | 25.76 |
| Agent 3-Hop | 0.529 | 0.648 | 0.511 | 0.884 | 26.75 |
| Agent 4-Hop | 0.536 | 0.657 | 0.525 | 0.880 | 27.19 |
| SemEval | | | | |
| | **A** | **$F_1$** | **R** | **P** | **|Nodes|** |
| BFS 1-Hop | 0.665 | 0.508 | 0.345 | 0.968 | 35.14 |
| BFS 2-Hop | 0.815 | 0.787 | 0.678 | 0.937 | 1565.20 |
| BFS 3-Hop | 0.751 | 0.754 | 0.759 | 0.750 | 3686.83 |
| BFS 4-Hop | 0.751 | 0.754 | 0.759 | 0.750 | 3843.54 |
| UnifiedQA-v2 | 0.497 | 0.653 | 0.943 | 0.500 | – |
| UnifiedQA-v2-T | 0.503 | 0.659 | 0.954 | 0.503 | – |
| UnifiedQA-v2-P | 0.566 | 0.651 | 0.805 | 0.547 | – |
| Agent 1-Hop | 0.647 | 0.460 | 0.299 | 1.000 | 13.43 |
| Agent 2-Hop | 0.769 | 0.714 | 0.575 | 0.943 | 26.83 |
| Agent 3-Hop | 0.775 | 0.727 | 0.598 | 0.929 | 28.60 |
| Agent 4-Hop | 0.751 | 0.699 | 0.575 | 0.893 | 29.09 |

around 50%. Moreover, in contrast, due to CauseNet, we can provide a source for each causal edge encountered on a path.

Moreover, as described in Section 4.1, we evaluated the effectiveness of providing triples (UnifiedQA-v2-T) from CauseNet as additional context to the language model as well as providing paths found by our agent as additional context to a language model (UnifiedQA-v2-P). The results indicate that UnifiedQA-v2-T slightly outperforms the vanilla UnifiedQA-v2 without context as well as UnifiedQA-v2-T with triples.

## 4.3   Ablation Study

In our ablation study in Table 3, we investigate the performance impact of the components of our approach. We try out the following configurations: (1) without supervised learning, i.e., we run Algorithm 1 directly and train the policy and value network with policy gradients from scratch, (2) without Actor-Critic, we remove the critic and only run the REINFORCE algorithm, (3) we remove

**Table 3: Results of the ablation study, where we compare different configurations of our approach by removing (–) different components. The evaluation measures are abbreviated as follows: accuracy: A, $F_1$-Score: $F_1$, recall: R, precision: P.**

|  | MS MARCO | | | | SemEval | | | |
|---|---|---|---|---|---|---|---|---|
|  | A | $F_1$ | R | P | A | $F_1$ | R | P |
| Agent 2-Hop | **0.460** | **0.562** | **0.408** | 0.901 | **0.769** | **0.714** | **0.575** | 0.943 |
| − Beam Search | 0.293 | 0.306 | 0.184 | **0.911** | 0.613 | 0.374 | 0.230 | **1.000** |
| − Supervised Learn. | 0.342 | 0.397 | 0.257 | 0.891 | 0.682 | 0.538 | 0.369 | **1.000** |
| − Actor-Critic | 0.441 | 0.539 | 0.386 | 0.896 | 0.740 | 0.657 | 0.494 | 0.977 |
| − Inverse Edges | 0.422 | 0.513 | 0.359 | 0.899 | 0.740 | 0.651 | 0.483 | **1.000** |
| − LSTM | 0.426 | 0.518 | 0.363 | 0.900 | 0.738 | 0.646 | 0.475 | **1.000** |

the beam search and use greedy decoding to only sample the most probable path, (4) without inverse edges in the graph, (5) without LSTM only using the two feedforward networks.

Overall, beam search has the biggest impact on performance. When beam search is exchanged for greedy decoding, the accuracy drops from 0.460 to 0.293 on MS MARCO and from 0.769 to 0.613 on SemEval. Notably, greedy decoding slightly increases the precision on MS MARCO and does reach a precision of 1.0 compared to the 0.943 of beam search on SemEval. Thus, the number of false positives decreases when only using the most probable path. Supervised learning has the second highest impact and we analyze it in more detail below. Next, the Actor-Critic algorithm only has a minor impact with a difference of around 0.02-0.03 points accuracy on both datasets. Similarly, the removal of inverse edges results in a slight decrease in overall performance but an increase in precision on the SemEval dataset to 1.0. That is because the removal of inverse edges reduces the probability of finding false positives, as discussed in Section 5. Using a feedforward network instead of an LSTM slightly decreases peformance, too. We attribute this to the fact that a pure feedforward network can only encode a single node whereas an LSTM can capture a broader context of a node, namely the previously visited nodes on the path. In future work, inspired by Almasan et al. [2], we would like to capture an even larger context of nodes (e.g., 1-hop or 2-hop neighborhood) by employing a graph neural network (GNN).

### 4.4 Effects of Supervised Learning

We compare the performance of the agent when using different numbers of supervised training steps. Figure 2 shows the accuracy of the agent on the SemEval [37] test set depending on the number of reinforcement learning training steps. Each of the three runs was bootstrapped with a different number of supervised training steps. Thus, at step 0, we can see the accuracy directly after supervised learning without any training via reinforcement learning. We observe that the run with 100 steps is significantly worse than the runs with 200 and 300 steps. It starts at around 0.67 directly after supervised learning and increases to around 0.72. Whereas the difference between 200 to 300 steps is already a lot smaller. Both start between 0.72 and 0.73 and follow similar trajectories afterward to reach an accuracy of around 0.76 after 2000 reinforcement learning steps. To maintain the clarity of the figures, we did not

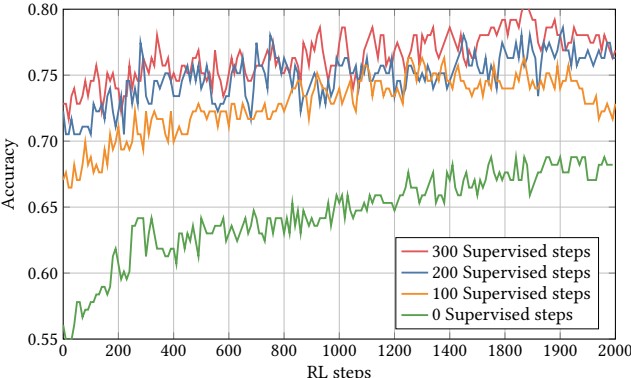

**Figure 2: Accuracy of the agent on the SemEval test set depending on the number of reinforcement learning training steps. Each run was bootstrapped with a different number of supervised training steps. Step 0 shows the performance directly after supervised learning.**

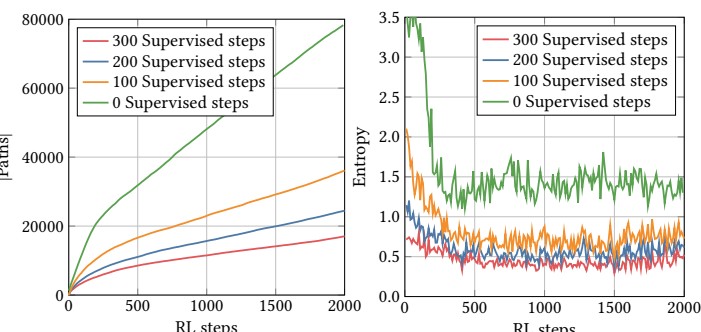

**Figure 3: Number of unique paths explored during reinforcement learning training on the left and the mean entropy of the action distribution of the policy network on the right. Each run was bootstrapped with a different number of supervised training steps.**

include a run with 400 steps, but the trend of diminishing returns on the number of supervised steps continues. This suggests that increasing the number of supervised steps beyond 300 does not improve performance. Likewise, we observed similar results on the MS MARCO [27] dataset.

Moreover, Figure 3 illustrates the number of unique paths explored during training on the left and the mean entropy of the action distribution of the policy network on the right. Notably, with an increasing number of supervised steps, the entropy of the policy network drops significantly. Hence, the number of explored paths during reinforcement learning also decreases as shown on the left in Figure 3. These findings indicate that supervised learning effectively establishes a strong foundation for the reinforcement learning agent. For example, an agent trained with 300 supervised steps only explores 21.6% of the paths of an agent without any supervised steps at the start. Accordingly, the agent trained with 300 supervised steps can exploit the knowledge acquired during

**Table 4: Some examples that our agent found. Each example consists of a cause, an effect, and a path between them. For each path, we indicate whether the agent used a normal *cause* edge, the *STAY* action, or an inverse edge *cause*$^{-1}$.**

---

**Cause:** h. pylori    **Effect:** vomiting

**Path:** h. pylori $\overset{cause}{\Longrightarrow}$ peptic ulcer disease $\overset{cause}{\Longrightarrow}$ vomiting $\overset{STAY}{\Longrightarrow}$ vomiting

---

**Cause:** Xanax    **Effect:** hiccups

**Path:** Xanax $\overset{cause}{\Longrightarrow}$ anxiety $\overset{cause}{\Longrightarrow}$ stress $\overset{cause}{\Longrightarrow}$ hiccups

---

**Cause:** chocolate    **Effect:** constipation

**Path:** chocolate $\overset{cause}{\Longrightarrow}$ constipation $\overset{cause}{\Longrightarrow}$ depression $\overset{cause^{-1}}{\Longrightarrow}$ constipation

---

supervised learning to follow better paths. In contrast, the agent without any supervised learning steps requires more exploration and fails to achieve the same performance, as shown in Table 3.

### 4.5 Examples

In Table 4, we illustrate a few example paths found by our agent. The paths can be used to follow the complete reasoning chain to examine the mechanisms of how a cause produces an effect. For example, the first path indicates that the bacterium *helicobacter pylori* can lead to the development of *peptic ulcer* disease, which can lead to *vomiting*. This example also demonstrates the agent's ability to utilize the *STAY* action.

Moreover, the agent learned to use inverse edges to recover from mistakes. In the third example, the agent went one step too far, from *constipation* to *depression*, and used an inverse edge afterward to return to the *constipation* entity.

### 5 DISCUSSION

*Inverse Edges.* For CauseNet [13], the addition of inverse edges implies that the agent can also walk from an effect to its cause. In general, their addition has a few benefits, like the possibility to undo wrong actions and to reach nodes that otherwise could not be reached under a given path length constraint. Conversely, they also introduce the possibility to make mistakes through false positives. While these mistakes will always happen for the BFS, our agent can minimize them by pruning the search space, as demonstrated in our experiments in Section 4.2.

Moreover, it might not always be clear that it makes sense to take an inverse edge in theory. For example, given a question like *"Does X cause Y?"*, we start the search at *X*. If we take an inverse edge from an entity *Z* to *Y* at some point during the search, it does not directly follow that *X* causes *Y* in theory. However, as our experiments show (Section 4.3), adding inverse edges improves performance in practice, so their benefits seem to outweigh the pitfalls. This is in line with prior works that also added inverse edges to improve performance [6, 48].

*Additional Causal Question Types.* At the moment, our approach only supports binary causal questions. In the following, we discuss a straightforward extension to open-ended questions. For example,

questions like *"What causes X?"* or *"What are the effects of X?"*, i.e., questions that ask for the cause of a given effect or vice versa.

Contrary to binary causal questions, we no longer have to verify a given causal relation but search for a suitable entity. Therefore, we need to change the decoding at inference time since we no longer know the entity we are looking for. Thus, we can introduce a majority voting approach. Given a question like *"What causes pneumonia?"*, we still sample multiple paths from the agent. Next, we count the occurrence of the entities as endpoints on these paths and select the one with the highest count as the answer. Alternatively, we can conduct the ranking by summing their probabilities on different paths. Moreover, we can also provide multiple answers by selecting the top-n candidates. Overall, this approach only requires minor changes to our current code base. To answer more complex questions like *"How does smoking cause cancer?"*, we can run the agent to answer the question as if it were a binary question and then use the paths and provenance data as explanations.

*Explainability of Causal Questions.* Our approach has the advantage of not only being able to answer binary causal questions but also providing explanations in the form of paths. Of course, this only holds for questions that were answered with "yes" because CauseNet [13] does not contain negative information.

As the examples in Section 4.5 show, the paths found during inference can be used to follow the complete reasoning chain. Thus, we can inspect the paths to examine the specific mechanisms by which the cause produces the effect, potentially through a chain of multiple entities. Moreover, since the agent might find multiple paths between a cause and effect pair, we can showcase multiple ways in which the cause and effect are related.

Finally, we can use the additional provenance data that is part of each relation [13]. For example, we can reference the original sentence from which the relation was extracted, which may provide additional insights. Furthermore, each relation contains the URL of the original web source, which can be checked to verify the causal relations and receive further information.

### 6 CONCLUSION

In this paper, we propose the first reinforcement learning approach for answering binary causal questions on knowledge graphs. Given a question "Does X cause Y?", we model the problem of finding causal paths as a sequential decision process over the causality graph CauseNet. We evaluate our approach on two causal question answering datasets. The results show that our reinforcement learning agent efficiently prunes the search space. Unlike language model-based approaches, our graph-approach with CauseNet yields high-precision and verifiable answers: for each single edge on a path, we can provide its original source on the web. In future work, we will extend the approach to open-ended causal question answering, as discussed in Section 5.

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

**Table 5: Statistics of CauseNet, including its precision, number of entities, and number of relations as reported by Heindorf et al. [13].**

| Graph | Precision | |Entities| | |Relations| |
|---|---|---|---|
| CauseNet | 96% | 80,223 | 197,806 |

## 7 APPENDICES

In this appendix, we give further details on the causality graph CauseNet that our reinforcement learning agent traverses and how we constructed the dataset of causal questions. Moreover, we give additional details on how we train our reinforcement learning agent. Finally, we report additional experiments regarding the beam width in the decoding phase.

### 7.1 CauseNet

CauseNet [13] is a large-scale causal knowledge graph of causal relations that were extracted from web sources like Wikipedia and ClueWeb12[9]. Formally, CauseNet is defined as $\mathcal{K} = (\mathcal{E}, \mathcal{R})$, where $\mathcal{E}$ represents the set of entities and $\mathcal{R}$ is the set of relations. The entities in CauseNet are single words or noun phrases, while the set of relations contains only the *mayCause* relation, i.e., $\mathcal{R} = \{mayCause\}$. Figure 1 shows an excerpt from CauseNet. It shows the entity *pneumonia* together with some of its causes and effects. The causal relations are stored as triples, e.g., $(pneumonia, mayCause, sepsis) \in \mathcal{K}$ where $pneumonia, sepsis \in \mathcal{E}$. Moreover, CauseNet contains additional meta-information for each relation. Included are the original source, i.e., the URL of the web page and the original sentence, if applicable.

### 7.2 QA Dataset Construction

To construct a dataset of binary causal questions, we extracted them from the Webis-CausalQA-22 corpus [3].[10] Webis-CausalQA-22 is a corpus for causal question answering containing around 1.1 million causal questions. The corpus was constructed by extracting causal questions from ten well-known question answering datasets including SQuAD v2.0 [33] and MS MARCO [27]. Other datasets have few causal questions to begin with, e.g., NewsQA [43] and HotpotQA [49], or are focused on open-ended question answering, e.g., ELI5 [9].

To extract binary causal questions, we build on work by Heindorf et al. [13] who extract questions via patterns of the form

`[question word]? [cause/effect] [cue word] [cause/effect]`

where the `[question word]` placeholder either represents one of the question words from Table 7 or is empty. The `[cue word]` placeholder represents words that are good indicators for causal relations together with their appropriate prepositions. The original approach only considers *cause* in different verb forms, e.g., infinitive, past, or progressive. We extend this to a greater number of causal cue words. Specifically, we use the collection from Girju and Moldovan [10] who curated a collection of causal cue words and ranked them by their frequency and ambiguity, i.e., how often they

---

[9]https://lemurproject.org/clueweb12/
[10]https://zenodo.org/record/7476615

**Table 6: Causal cue words for detecting binary causal questions. The list was curated by Girju and Moldovan [10] and includes words that frequently indicate causality while having low ambiguity.**

| Causal Cue Words | | | |
|---|---|---|---|
| induce | provoke | relate (to) | trigger off |
| give rise (to) | arouse | link (to) | bring on |
| produce | elicit | stem (from) | result (from) |
| generate | lead (to) | originate | trigger |
| effect | derive (from) | bring forth | cause |
| bring about | associate (with) | lead up | |

**Table 7: Question words used for binary question extraction (left) and POS tags used for exclusion (right).**

| Question Words | | POS-Tag | Description |
|---|---|---|---|
| is | do | CC | Coordinating conjunction |
| can | does | IN | Preposition or subordinating conj. |
| might | did | TO | To-prepositions |
| would | will | WDT | Wh-determiner |
| could | are | WP | Wh-pronoun |
| may | | WRB | Wh-adverb |

appear in text and how often they refer to a causal relation. Among these, we selected the ones that were ranked with high frequency and low ambiguity. Table 6 shows the full list of 23 words.

Moreover, the `[cause/effect]` placeholder represents causal concepts, where one takes the role of the cause and the other the role of the effect. The order depends on the question word and the causal cue word. As done by Heindorf et al. [13], we place a few restrictions on the questions to keep the concepts simple and increase the probability that they can be found in CauseNet. The restrictions are enforced by filtering questions based on POS-Tags from the Stanford CoreNLP [25], e.g., we disallow coordinating conjunctions and subordinating conjunctions. The full list is shown in Table 7 on the right. Finally, we check whether the questions are answered with "yes" or "no" and remove any further explanation.

### 7.3 Pseudocode for Training the Reinforcement Learning Agent

In the following, we describe the training procedure of our reinforcement learning agent. This includes the pre-processing of the questions, the sampling of path rollouts, and the update rules for the weights of the agent. We start with the observation that CauseNet [13] does not contain negative information. Thus, we only train the agent on positive causal questions, i.e., questions whose answer is "yes". Similarly, we must remove all questions where the cause, effect, or both cannot be found in CauseNet.

Algorithm 1 displays the pseudocode of the whole training phase. The pseudocode assumes that only positive causal questions, where cause and effect can be found in CauseNet, remain in the given *questions*. First, we pre-process the questions by linking the cause and effect to the corresponding entities $e_c$ and $e_e$ in CauseNet. This

**1 Input:** Knowledge graph $\mathcal{K}$, questions *questions*,
optimization steps *steps*, batch size $B$, path rollout length
$T$, learning rate $lr$, entropy weight $\beta$, discount factor $\gamma$,
GAE lambda $\lambda$

**2 Output:** Trained Agent $\theta, \psi$

**3 Function** Training($\mathcal{K}$, *questions*, *steps*, $B$, $T$, $lr$, $\beta$, $\gamma$, $\lambda$)**:**

  **4**    *processed_questions* = [ ]

  **5**    **for each** $q$ **in** *questions* **do**

  **6**      Link the cause and effect of $q$ to entities $e_c, e_e$ in $\mathcal{K}$

  **7**      Compute embeddings $\mathbf{q}, \mathbf{e_c}$ for $q, e_c$

  **8**      *processed_questions.append*$((\mathbf{q}, e_c, \mathbf{e_c}, e_e))$

  **9**    **end**

  **10**

  **11**    Initialize agent weights $\theta, \psi$

  **12**    *path_rollouts* = [ ]

  **13**    *step* = 0

  **14**    **while** *step* < *steps* **do**

  **15**      *path_rollout* = [ ]

  **16**      $(\mathbf{q}, e_c, \mathbf{e_c}, e_e)$ =
       $SampleUniform(processed\_questions)$

  **17**      $s_0 = (\mathbf{q}, e_c, \mathbf{e_c}, \mathbf{0}, e_e)$

  **18**      **for** $t = 0$ **to** $T - 1$ **do**

  **19**        $a_t = SampleCategorical(\pi_\theta(a_t|s_t))$

  **20**        Receive state $s_{t+1} = \delta(s_t, a_t)$ and reward
         $r_t = \mathcal{R}(s_{t+1})$

  **21**        *path_rollout.append*$((s_t, a_t, r_t))$

  **22**      **end**

  **23**

  **24**      *path_rollouts.append*(*path_rollout*)

  **25**      **if** |*path_rollouts*| = $B$ **then**

  **26**        Compute the GAE $\mathcal{A}_t^\psi$ and $\lambda$-returns $R_t(\lambda)$ for
        each episode in *path_rollouts* using $\lambda, \gamma$

  **27**        *policy_update*, *value_update* = 0

  **28**        **for each** $((s_0, a_0, r_0, \mathcal{A}_0^\psi, R_0(\lambda)), \ldots,$

  **29**        $(s_{T-2}, a_{T-2}, r_{T-2}, \mathcal{A}_{T-2}^\psi, R_{T-2}(\lambda)))$ **in**
        *path_rollouts* **do**

  **30**          *policy_update* +=
          $\sum_{t=0}^{T-2} \nabla_\theta \log \pi_\theta(a_t|s_t) \mathcal{A}_t^\psi$

  **31**          *value_update* += $\sum_{t=0}^{T-2} \nabla_\psi (R_t(\lambda) - V_\psi(s_t))^2$

  **32**        **end**

  **33**        Compute the entropy regularization term $H_{\pi_\theta}$

  **34**        *policy_update* = $-\frac{policy\_update}{|path\_rollouts|}$,
        *value_update* = $\frac{value\_update}{|path\_rollouts| \cdot (T-1)}$

  **35**        $\theta = \theta - lr \cdot (policy\_update + \beta H_{\pi_\theta})$

  **36**        $\psi = \psi - lr \cdot value\_update$

  **37**        *path_rollouts* = [ ]

  **38**        *step* = *step* + 1

  **39**      **end**

  **40**    **end**

**41 return** $\theta$

**Algorithm 1:** Pseudocode of the training procedure for the
policy network $\pi_\theta(a_t|s_t)$ and value network $V_\psi(s_t)$.

is followed by the computation of embeddings for the question and
entities.[11] Next, the weights $\theta$ and $\psi$ of the agent are initialized.
In the default setup, both are initialized randomly. In the case of
a preceding supervised learning phase (Section 3.6), the weights
of the policy network $\theta$ are initialized with the resulting weights
from the supervised learning.

Afterward, we start sampling path rollouts from the environment
via the current policy network $\pi_\theta(a_t|s_t)$. Each path rollout has the
same length $T$. Hence, the agent should learn to use the *STAY* action
in case it arrives at the target entity before a length of $T$ is reached.
Given a pre-processed question $q$, we construct the first state $s_0$.
Subsequently, the agent interacts with the environment for $T$ time
steps. At each time step, the agent applies an action $a_t$ and receives a
reward $r_t$ while the environment evolves via the transition function
$\delta(s_t, a_t)$ to the next state $s_{t+1}$. This procedure is continued until a
full batch of path rollouts is accumulated.

The training of the agent is facilitated via the Synchronous Advantage Actor-Critic (A2C) [26] algorithm. The policy network
$\pi_\theta(a_t|s_t)$ takes the role of the actor while the value network $V_\theta(s_t)$
takes the role of the critic. We briefly experimented with Proximal
Policy Optimization (PPO) [36] but found no significant performance improvements. Thus, the update rule for the policy network
$\pi_\theta(a_t|s_t)$ becomes

$$\nabla_\theta J(\theta) = -\frac{1}{B} \sum_i^B \sum_{t=0}^{T-2} \nabla_\theta \log(\pi_\theta(a_t|s_t)) \, \mathcal{A}_t^\psi \quad (6)$$

where $B$ is the batch size, T the path rollout length, and $\mathcal{A}_t^\psi$ the
generalized advantage estimate (GAE) [35]. GAE introduces two
hyperparameters, the discount factor $\gamma$ and a smoothing factor $\lambda$
which controls the trade-off between bias and variance [35, 42].

As commonly done, we add an entropy regularization term to
the objective [6, 18]. The entropy regularization should help the
agent with the exploitation vs. exploration tradeoff. Specifically,
it should encourage exploration during training and the resulting
policy should be more robust and have a higher diversity of explored
actions. We compute the average entropy of the action distribution
of $\pi_\theta(a_t|s_t)$ over all actions $a_t \in \mathcal{A}(s_t)$ at each time step $t$ and take
the average over the whole batch:

$$H_{\pi_\theta} = \frac{1}{B(T-1)} \sum_i^B \sum_{t=0}^{T-2} (-\sum_{a_t \in \mathcal{A}(s_t)} \pi_\theta(a_t|s_t) \log \pi_\theta(a_t|s_t)) \quad (7)$$

The final update for the policy network becomes

$$\theta = \theta - lr \cdot (\nabla_\theta J(\theta) + \beta H_{\pi_\theta}) \quad (8)$$

where $lr$ is the learning rate and $\beta$ is a hyperparameter that determines the weight of the entropy regularization term.

Simultaneously, we update the value network $V_\psi(s_t)$ via the
mean-squared error between the $\lambda$-return and the predictions of
the value network

$$\nabla_\psi J(\psi) = \frac{1}{B(T-1)} \sum_i^B \sum_{t=0}^{T-2} \nabla_\psi (R_t(\lambda) - V_\psi(s_t))^2 \quad (9)$$

---

[11]We use GloVe [30] embeddings to embed the questions and entities. For more details,
see Section 4.1.

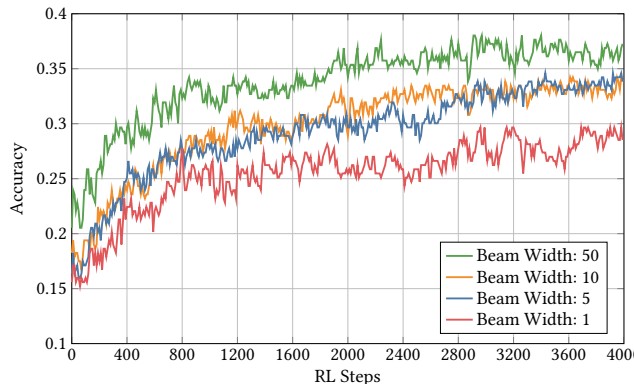

**Figure 4: Accuracy on the MS MARCO test set depending on the number of reinforcement learning training steps. Each run uses a different beam width.**

where $R_t(\lambda)$ is the $\lambda$-return [29, 42]. Therefore, the update for the value network becomes:

$$\psi = \psi - lr \cdot \nabla_\psi J(\psi) \qquad (10)$$

### 7.4 Decoding Analysis

In the following, we investigate the effects of different beam widths on the MS MARCO test set. We experiment with beam widths of 1, 5, 10, and 50 and present the results in Figure 4. As the beam width increases, accuracy also increases. For example, the difference between a width of 1 and a width of 50 is 0.09 points accuracy after 2000 steps. In general, beam search can be viewed as an interpolation between greedy decoding and BFS. If the beam width is set high, the agent performs close to an exhaustive search. We can already observe this effect when looking at step 0. As the beam width increases, the accuracy without any learning also increases slightly, e.g., for widths of 10 and 50 from 0.18 to 0.21. Having said this, even beam width 50 only starts at 0.21 and still has a lot of learning progress afterward and the performance of width 50 does not reach the performance of supervised learning suggesting that 50 is still a reasonable beam width.

