# OpenReview forum: "Causal Question Answering with Reinforcement Learning"
_ACM.org/TheWebConf/2024/Conference — TheWebConf24 Oral_

### Official Review · Reviewer_fEbv · 2023-11-19

**Novelty:** 3
**Technical Quality:** 3

**Review:**

Summary:

The paper applies reinforcement learning on CauseNet for causal question answering and provides explanations or evidence for answers. The paper introduces a new causal QA dataset. The technique details are described clearly in the paper. It would be better for the authors to assign a name to their new dataset and appropriately cite it in the paper. The main drawback of the paper lies in the experimental part. From the table of main results, it is rather difficult to check how good the overall performance of the proposed method is and draw a conclusion. I highly appreciate that the authors provide the performance of GPT4 during the rebuttal. The experiments show that GPT4 performs very well (better than the baselines and the proposed method at most metrics). Based on the experimental results and the overall quality of the paper, I re-considered and decided to maintain the scores.

Pros:
+ The paper represents the pioneering effort in incorporating a reinforcement learning approach for addressing causal question answering on knowledge graphs.
+ The paper introduces a novel dataset specifically designed for causal questions.

Cons:
- The paper does not include a comparison with GPT models, but it is suggested that such a comparison could be conducted, especially given that the answers in the QA dataset are binary (Yes or No), making the comparison potentially less challenging.
- Additionally, more information about the dataset, such as the average question length and the average number of entities in each question, would be beneficial for understanding the complexity of the questions.

**Questions:**

* In Table 1, it would be better for there to be a total number of both datasets in each column.
* The description of QA dataset construction is different between Sec.4.1 and Sec.7.2. In Sec.4.1, the authors employ subsets of causal questions from two datasets but in Sec.7.2, the authors extracted the questions from Webis-CausalQA-22 that does not contain SemEval. Or did I miss some points? Hope the authors can explain.
* Can the authors use GPT (3.5 or 4) as a baseline to perform on the test set and provide the performance comparison?
* Why do the authors use GloVe as the pre-trained embeddings which is quite old? Is there any other better choice? Since there are lots of medical terminologies in the dataset, why don't use the domain-specific embeddings or models, e.g. BioBERT, ClinicalBERT, etc as the pre-trained model?

**Reviewer Confidence:**

3: The reviewer is confident but not certain that the evaluation is correct

**Scope:**

4: The work is relevant to the Web and to the track, and is of broad interest to the community

---

### Official Review · Reviewer_h2Lx · 2023-11-20

**Novelty:** 5
**Technical Quality:** 4

**Review:**

This paper takes an interesting approach to the problem of binary causal question answering - which is it tries to use a reinforcement learning approach to answer questions over a causal knowledge graph (CauseNet).  The approach is definitely useful, it has the advantage of generating answers that lead to explainable results and as far as I know, the approach is novel.  Brute force BFS ends up searching far more nodes than the RL approach - so the approach seems quite novel and useful.  Having said that, I think there are many different aspects of the current version that could be strengthened.  I do think the paper can provide an interesting contribution to the field but I think it is not yet ready for publication in a top tier venue in its current form.  All my concerns are described as questions below.

**Questions:**

1.  Why was the scope for QA so severely constrained to just binary causal question answering?  Surely the technique can be broadened to any binary QA or causal QA or QA in general?
2. The number of datasets used for evaluation is a bit underwhelming.  Why wasnt BoolQ (https://arxiv.org/abs/1905.10044) for instance considered? Also for MSMarco, could one not add more questions to make it more balanced and see if the performance changes for large language models?
3.  In the age of chatGPT and LLMs, it seems a bit limiting to test only with a single LLM.  Here's an example response from a version of mpt for one of the questions in the paper:
Does Xanax cause hiccups?  Answer yes or no.  If yes, explain.  If no, explain why.
Xanax is a medication used to treat anxiety disorders.  It is a benzodiazepine, which is a class of drugs that also includes Valium and Ativan.  Benzodiazepines are known to cause hiccups.  The exact mechanism is not known, but it is thought to be due to the way they affect the brain.
I realize this is a one-off example and that may not be correct even, but surely this points to a need for more comprehensive analysis here to make the case that the RL approach proposed here provides better, more explainable results consistently.
4. The results were a bit underwhelming as well - even on SemEval, the difference between LLM and the RL approach was 4% for the 4 hop case (assuming more hops should be better).  The point about MSMarco is well taken, but again this needs to be offset by more datasets.

**Reviewer Confidence:**

3: The reviewer is confident but not certain that the evaluation is correct

**Scope:**

4: The work is relevant to the Web and to the track, and is of broad interest to the community

---

### Official Review · Reviewer_Mg2e · 2023-11-23

**Novelty:** 6
**Technical Quality:** 6

**Review:**

The authors present an approach to causal question answering using a RL
framework applied to a large-scale dataset of causal relations,
CauseNet​​​​. The authors aim to address the limitations of current causal
question-answering methods that often lack explanations or evidence
supporting their answers. They propose an Actor-Critic based RL agent,
which is bootstrapped with supervised learning to manage large action
spaces and sparse rewards.


Pros
======

  + This paper introduces CauseNet, a dataset of causal relations with
    provenance, that provides a solid foundation for causal reasoning
    research.
  + The authors propose an Actor-Critic based RL agent, which is
    bootstrapped with supervised learning to manage large action spaces
    and sparse rewards. The approach is designed to prune the search
    space efficiently, demonstrating significant improvements over
    traditional methods like breadth-first search in terms of nodes
    visited per question​​.
  + The authors address a gap in the field of causal question answering
    by leveraging reinforcement learning and a large-scale causal
    relation dataset.
  + The authors present a clear and detailed methodology, effectively
    combining reinforcement learning with supervised learning for
    initial bootstrapping. The sequential decision-making framework for
    navigating the causal knowledge graph is a notable strength and a
    new application.
  + The paper is well-written and nicely formatted.
  + The inclusion of an ablation study supports the hypothesis and adds
    depth to the experimentation section.
  + The running example of pneumonia diagnosis in the introduction and
    throughout the paper solidifies the significance of the paper.
  + The contributions of the paper are sound: RL for causal QA on KG, a
    supervised learning algorithm, and a new dataset.
  + The experiments and the results support the hypothesis to develop an
    RL approach for binary causal reasoning on KGs.  I think this will
    be of interest to the web community.
  + The reduction in the number of visited nodes expanded (99%!) is
    impressive.  I think the authors can highlight this in the
    discussion/conclusion as well.


Weaknesses
============

  - The current dataset only considers a single relation (the
    cause-and-effect relation), but more validating more complex
    question-answers would be insightful.  (See question 1)
  - There is only one final reward.  I'm not sure if this accurately
    reflects the medical diagnosis setup (where small diagnoses /
    treatments may indicate incremental rewards).
  - There is limited comparisons with existing QA models.  I think
    including comparisons with more recent and advanced
    question-answering models might provide a better understanding of
    the RL agent's performance.

Rebuttal Update
============

After reading the author's comments, I have revised my technical quality score.

**Questions:**

1. I wonder how the current approach would scale on other types of KG
   relations?
2. What would training on negative and positive information look like?
3. I appreciate the footnote 4 about the episodes of different lengths
   (and not using STOP). I'm wondering how the findings of this paper
   would look like with paths of different lengths?
4. I'm wondering how this approach would scale to negative examples and/or
   stochastic actions?
5. How was the learning rate found of 0.0001 found? (It seems that that
   was constant before the hyperparameter optimization).
6. How well does this method work when the amount of data or the
   complexity of questions increases significantly? Can it handle
   larger datasets without losing accuracy or speed?

**Reviewer Confidence:**

3: The reviewer is confident but not certain that the evaluation is correct

**Scope:**

4: The work is relevant to the Web and to the track, and is of broad interest to the community

---

### Official Review · Reviewer_ABPN · 2023-11-23

**Novelty:** 4
**Technical Quality:** 5

**Review:**

This paper proposes to use reinforcement learning methods for performing explanatory binary causal question answering. The main contributions of the paper are centred around the use of knowledge graphs coupled with reinforcement learning for deriving causal paths that can be used for understanding how particular YES decisions are found. In general, the article is well-written and the approach is well-explained. From the description of the algorithms, it is not totally clear how different is the proposed approach compared to previous works besides the use of the CauseNet knowledge graph. One important discussion missing from the result comparison section and discussion section is the impact of not using NO causal relationships for pre-training the model and the fact that at inference time the approach defaults to a NO answer when no cause and effect can be found. This behaviour may lead to false negatives which is not discussed in the paper. Wouldn't it be more appropriate to return UNKNOWN when there is simply a lack of information in the knowledge graph? I believe the paper should include a discussion about the impact of the defaulting behaviour in relation to the false negatives (this may explain the low recall of the proposed model which is not discussed in the paper). The discussion of the result also does not appropriately discuss results besides the precision which tends to favour their models compared to the other metrics (for example the Accuracy of the proposed model is very low on MS MARCO compared to UnifiedQA) this lack of detailed discussion make the significant of the paper weaker.

Pros:
- The paper is well written and the running example helps to clarify the aims of the paper.
- The ablation study is thorough and adds some additional knowledge about the strength of the individual model components.
- The proposed reinforcement learning approach appears to have clear benefits when compared to the BFS baseline.

Cons:
- The impact of defaulting to NO answer is not discussed in the paper and how it may affect the results (in particular the false negatives).
- The authors focus their analysis on the precision and fail to acknowledge the other reported metrics such as the really poor accuracy of their model on MS MARCO and the evaluation only focuses on one alternative model.
- Path explanations are only valid for YES answers. There is no discussion about how future work could deal with such an issue.
- The novelty of the approach is unclear besides the use of CausalNet (most of the model appears to be heavily based on DeepPath, MINERVA and SRN).

--- Rebutal Update ---

After reading the author's comments, I have revised my novelty rating. Please make sure that the information provided in the comments is added to the paper.

**Questions:**

- How different is the proposed approach (besides the use of CausalNet) compared to DeepPath, MINERVA, SRN and others?
- Why do the proposed models have lower recall than the other models? Why is the accuracy much lower on MS MARCO?
- What is the impact of only pre-training on YES relationships? What is the impact of defaulting to a NO answer when no cause and effect can be found?
- What would be a good approach for dealing with NO explanation paths (future work)?

**Reviewer Confidence:**

3: The reviewer is confident but not certain that the evaluation is correct

**Scope:**

3: The work is somewhat relevant to the Web and to the track, and is of narrow interest to a sub-community

---

### Official Review · Reviewer_YH8s · 2023-11-25

**Novelty:** 5
**Technical Quality:** 4

**Review:**

The work introduces a reinforcement learning-based method for causal learning/Q&A. an Authors introduce an Actor-Critic based agent which learns to search through the graph to answer causal questions. Empirical results seems good. The key focus is- whether authors can model the causal question-answering task as a sequential decision problem over CauseNet.

The idea is fresh and novel. I can not recall on top of my head if I have read a similar idea recently that attempts to handle causal Q&A using reinforcement learning. I am also okay with proposed evaluation and results are satisfactory.

**Questions:**

Q1. The statement: As CauseNet [13] does not contain negative information (see Section 7.1 in appendix), we only train the agent on positive causal questions, i.e., questions whose answer is “yes”--> is it a limitation of the proposed work that it is too dataset specific?

Q2. I see a lot of mention of CausalNET, making the work too specific to a single Causal KG. My suggestion is trim down reference of CausalNET and make it more generic- something like causal KG. Currently paper looks like it is aiming to “crack” one KG. I do know this domain is new, but will grow over time. Hence, making the work too specific to one KG, specially in writing is not giving proper message.

Q3: For explainability, human evaluation is missing. Empirical evaluation normally is good as first step, but human evaluation normally makes the work solid and full-proof. Can author explain why human evaluation has been omitted from the empirical study?

Q4: What are the limitation of this work? What are concrete items which research community can learn from this work that are clear assumptions in this work, that can be relaxed in following up work (next work on this topic).

Q5. The system like  CONQUER [18] also does reinforcement learning, but in conversational setting. I do miss strong baselines in this paper. Would it make sense to include  CONQUER [18] as one of the baseline turning off its dialog history? BFS by heuristic will perform bad.

**Reviewer Confidence:**

3: The reviewer is confident but not certain that the evaluation is correct

**Scope:**

4: The work is relevant to the Web and to the track, and is of broad interest to the community

---

### Decision · Program_Chairs · 2024-01-22

**Decision:**

Accept (Oral)

**Comment:**

The reviewers agree that the paper is novel and original. There are points of concern including in particular ad-hocness to one KGs only (CauseNet), as well as suggestions for improvement . Yet there seems to be consensus on the strengths including solid experimental setting and interesting problem to tackle.